# A quality management system aiming to ensure regulatory-grade data quality in a glaucoma registry

Shinsuke Wada[1], Satoru Tsuda[2], Maiko Abe[2], Toru Nakazawa[2,3,4,5], Hisashi Urushihara[1]*

1 Division of Drug Development and Regulatory Science, Graduate School of Pharmaceutical Sciences, Keio University, Tokyo, Japan, 2 Department of Ophthalmology, Graduate School of Medicine, Tohoku University, Sendai, Japan, 3 Department of Retinal Disease Control, Tohoku University Graduate School of Medicine, Sendai, Japan, 4 Department of Ophthalmic Imaging and Information Analytics, Tohoku University Graduate School of Medicine, Sendai, Japan, 5 Department of Advanced Ophthalmic Medicine, Tohoku University Graduate School of Medicine, Sendai, Japan

* urushihara.hisashi@keio.jp

**Data Availability Statement:** All relevant data are within the paper and its Supporting information files.

**Funding:** The authors received no specific funding for this work.

## Abstract

### Background

Disease/patient registries are underutilized despite their multiple advantages over clinical trials in the clinical evaluation of drugs, such as the capacity for long-term curation, provision of patient outcome data in routine clinical practice, and provision of benchmark data for comparison. Ensuring the fit-for-purpose quality of data generated from such registries is important to informing regulatory decision making. Here, we report the construction of a quality management system aiming to ensure regulatory-grade data quality for a registry of Japanese patients with glaucoma to evaluate long-term patient outcomes.

### Methods

The quality management system was established by reference to the risk-based approach in the ICH-E6 (R2) recommendations. The following three-component approach was taken: establishment of governance, computerized system validation (CSV), and implementation of risk assessment and control. Compliance of the system with the recommendations of regulatory guidelines relevant to use of the registry was assessed.

### Results

Governance by academic collaboration was established. This was followed by the development of a total of 15 standard operating procedures, including CSV, data management, monitoring, audit, and management of imaging data. The data management system was constructed based on a data management plan, which specified data/paper flow and data management procedures. The electronic data capture (EDC) system was audited by an external vendor, and configured and validated using the V-model framework as recommended in the GAMP5 guideline. Informed consent, eligibility assessment and major ophthalmology measurements were determined as Critical to Quality (CTQ) factors. A total of

**Competing interests:** I have read the journal's policy and the authors of this manuscript have the following competing interests: SW is an employee of Pfizer R&D Japan, and SW is also a graduate student at Keio University and contributed to the present study independent of Pfizer R&D Japan. HU received research grants from Senju Pharmaceutical Co.,Ltd. The funder had no role in study design, data collection and analysis, decision to publish, or preparation of the manuscript. The other authors have declared that no competing interests exist. This does not alter our adherence to PLOS ONE policies on sharing data and materials.

22 risk items were identified and classified into three categories, and operationalized in the form of a risk control plan, which included training sessions and risk-based monitoring. The glaucoma registry addressed most quality recommendations in official guidelines issued by multiple health authorities, although two recommendations were not met.

## Conclusions

We established and configured a quality management system for a glaucoma registry to ensure fit-for-purpose data quality for regulatory use, and to curate long-term follow-up data of glaucoma patients in a prospective manner.

## Introduction

Disease/patient registries are powerful tools in the collection and curation of data on the natural history of diseases and patient journey. Unsurprisingly, these registries have recently attracted the attention of the regulatory community. The use of disease registries has informed regulatory decision making, especially in the context of evaluation of post-marketing efficacy and safety of medical products. Disease registries can also serve as an external control arm in clinical trials. Several official sets of guidance on the use of real-world data (RWD), including registry data, have been issued in the United States (US), European Union (EU), and Japan in the last decade [1–8].

Policies regarding the fit-for-purpose quality of clinical trial data and RWD in regulatory decision making have been discussed around the world. A notable example is the International Council for Harmonization (ICH) documents: the ICH-E6 guideline "Good Clinical Practice (GCP)" revision (R2), issued in November 2016, stipulates that sponsors should implement a system to manage data quality throughout all stages of a clinical trial, and recommends the introduction of a risk-based approach [9]. A reflection paper on GCP renovation stated that the coming ICH-E6 R3 will emphasize the use of RWD, including patient-reported outcomes (PRO), to support regulatory decision making for marketing authorization [10]. In addition, the ICH-E8 (R1) guideline stresses that fit-for-purpose quality should be considered in the use of RWD to support good regulatory decision making [11]. Thus, these various policies acknowledge the importance of disease/patient registries in regulatory use and indicate the importance of ensuring data quality in their development and maintenance. When defining data quality in disease registries, the two most cited quality attributes are completeness and accuracy [12].

Of note, use of disease registries for new drug applications (NDAs) have been limited to the cases for a few orphan drugs, including cerliponase alfa and defibrotide sodium [13, 14]. As long-term randomized controlled trials are not the design of choice for studies assessing PROs, disease registries may supplement experimental studies with long-term and/or PRO data collected in routine practice. Indeed, PRO measures are essential to incorporating the perceived value of a treatment even for common diseases from the patient's perspective in benefit/risk assessment. One disease area that might benefit from regulatory use of a disease registry is glaucoma. Glaucoma is a common ophthalmological disease which is characterized by degeneration of the optic nerve and subsequent visual impairment. It affects more than 76 million people around the world [15]. Both quality of vision (QOV) and quality of life (QOL) are impaired. Moreover, approximately 10% of patients experience bilateral blindness, making it the leading cause of irreversible blindness in the world [16, 17]. Emerging treatment modalities of neuroprotective agents targeting the optic nerve and retinal blood flow are attracting

attention [18–20]. Accordingly, the importance of collecting and evaluating long-term, multiple clinical and PRO data—including glaucoma visual field and blood flow—in clinical development for new glaucoma therapies is now firmly recognized.

Here, we report a quality management system constructed for a Japanese glaucoma registry (UMIN000037627) as a platform for collecting data on the patient journey with regulatory-grade quality. We also assessed the compliance of this system with recommendations in several regulatory guidelines on the use of RWD issued in the US, EU and Japan.

## Methods

### Defining the registry's goal and purpose

The glaucoma registry aimed to curate clinical test and PRO findings in patients with glaucoma and provide longitudinal data with regulatory-grade quality. Further, the registry data was intended to serve as a potential historical control for comparison with glaucoma patients treated with investigational drugs in future single-arm clinical trials.

### Establishing quality management system

The quality management system for the glaucoma registry was constructed based on a risk-based approach according to the recommendations of the ICH-E6 (R2) guideline, principally by the inclusion of three components: establishment of governance, computerized system validation (CSV), and implementation of risk assessment and control (Fig 1).

**Establishment of governance.**  In accordance with the Agency for Healthcare Research and Quality (AHRQ) "Registries for Evaluating Patient Outcomes: A User's Guide: 4th Edition", governance for the registry was defined as a formalized structure or plan for managing the registry and guiding decision making related to registry funding, operations, and dissemination of information [6]. We operationalized the governance framework in the form of an organization and standard operational procedures (SOPs) which were developed and implemented specifically for the registry. The protocol was developed to identify registry-specific requirements, including target subjects, sample size, ophthalmology measurements, and visiting schedule. The data management system and plan were developed to manage data items and the quality required for the glaucoma registry. The data management system consisted of an electronic data capture (EDC) system and imaging data repository.

**Computerized system validation.**  A computerized system, including the EDC system, was created for the registry to record and store the data curated for the glaucoma registry. It was validated by reference to the Good Automated Manufacturing Practice version 5 (GAMP5) guideline, the global standard of CSV for clinical trials in compliance with GCP [21]. The GAMP5 guideline recommends that the process of CSV starts with a vendor audit and that the computerized system should be validated through the V-model framework (Fig 2). The vendor audit was performed with reference to the GAMP5 guideline and the electronic records and electronic signatures (ER/ES) guideline; this was done because the EDC system would impact the quality and integrity of data collected in the registry, and the EDC system should therefore be reviewed against GCP standards as to whether it would allow the project goals to be achieved [21, 22].

**Risk assessment and control.**  In accordance with the stepwise procedures recommended in the ICH E6 (R2) guideline, risk assessment started with determining the "critical data and process" for a given study. This is a similar concept to the "critical to quality (CTQ) factors" proposed in the ICH E8 (R1) guideline [11]. CTQ factors are those data items and study processes which are essential to ensure the integrity and reliability of study conclusions and the protection of trial participants. Risk is defined as errors and biases which potentially jeopardize

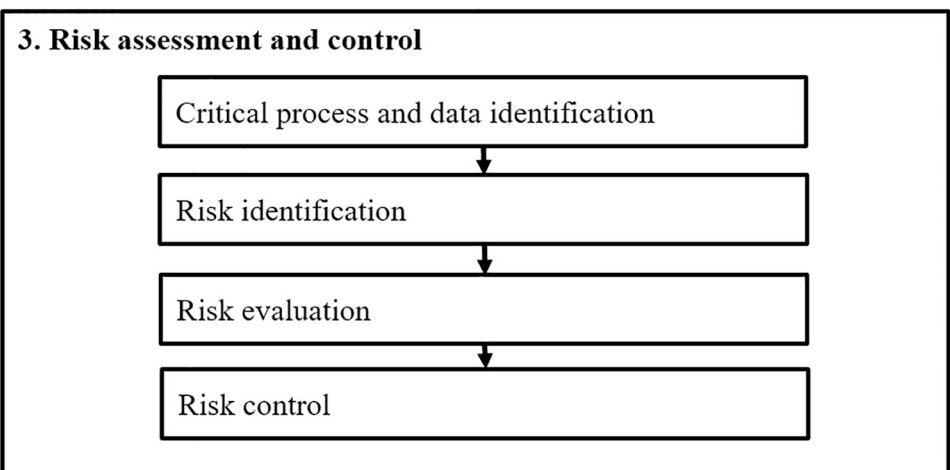

**Fig 1. Construction of a quality management system for the glaucoma registry.** Quality management system for the glaucoma registry consisted of 3 components: establishment of governance, computerized system validation, and risk assessment and control.

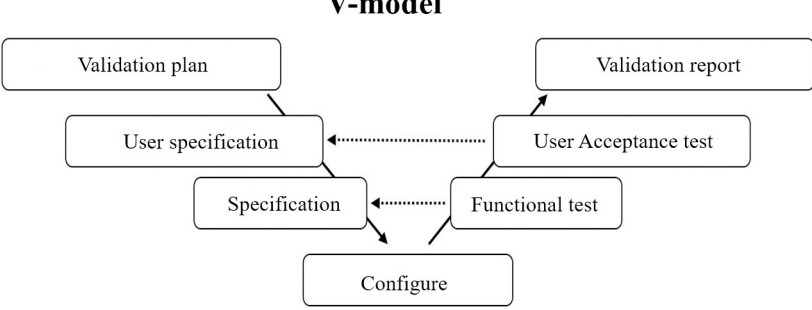

**Fig 2. V-model framework for computerized system validation.** The V-model is a known validation process framework for computerized systems, and is recommended in GAMP 5. The left side of the "V" represents the decomposition of requirements and creation of system specifications. The right side of the "V" represents the integration of parts and their validation, which consists of functional testing and user acceptance testing (UAT).

the CTQ factors. After the risk items for the present registry were identified, each was subsequently graded as high, middle, or low in terms of probability (i.e. likelihood of errors occurring), impact (i.e. impact of such errors on human subject protection and the reliability of results), and detectability (i.e. extent to which such errors would be detectable), in accordance with the Risk Assessment Categorization Tool (RACT) recommendation for clinical trials by TransCelerate [23]. For each risk item, a total risk score was calculated and used to classify the risk item as high (10 to 27 points), medium (4 to 9 points), or low risk (1 to 3 points).

The risk control plan in this registry included training and risk-based monitoring. The latter is an adaptive approach that directs monitoring focus and activities to those areas having the greatest potential to impact subject safety and data quality. The study protocol and monitoring plan specified the frequency and interval of on-site and/or central (remote) monitoring, and determined the required intensity of monitoring for each risk item in accordance with its risk score; if the risk item was classified as high risk, intensive monitoring was required, including source data verification (SDV) for all relevant data as specified in the study monitoring plan. These risk items were also reflected in development of the training programs and configuration of automated edits of the EDC, after the consideration of weighting by their scores.

## Compliance with the recommendations of regulatory guidelines

Compliance of the quality management system in the glaucoma registry with the following regulatory guidelines relevant to the use of RWD was assessed: 1) US Food and Drug Administration (FDA), Use of Real-World Evidence to Support Regulatory Decision-Making for Medical Devices [4]; 2) European Medicines Agency (EMA), Discussion paper: Use of patient disease registries for regulatory purposes–methodological and operational considerations [5]; 3) International Medical Device Regulators Forum (IMDRF), Tools for Assessing the Usability of Registries in Support of Regulatory Decision-Making [24]; and 4) Japanese Ministry of Health, Labour and Welfare (MHLW), Points to Consider for Ensuring the Reliability in Utilization of Registry Data for Applications [8]. These guidelines relate to the data quality and integrity of patient registries and were the currently available regulatory documents as of August 2022. The quality management system in the glaucoma registry was evaluated against each recommendation in each of these guidelines.

## Results

### Quality management system

**Establishment of governance.** The glaucoma patient registry project was initiated in April 2018 (Fig 3). An organization to control the governance and management of the registry was established by collaboration between Tohoku University and Keio University. Considering a small number of study sites, the system owner was fully responsible for supervising the whole project of the present registry, and we sought a simple governance structure without establishing the steering committee as suggested by AHRQ guidance [6]. Almost all the foreseeable operations of the registry including change control have already been operationalized in the form of the SOPs and manuals. Outstanding issues and unpredictable changes in the operations and surrounding environment of the registry were to be discussed among the system owner, study manager, data manager and CSV manager on ad hoc basis, if applicable. An opportunity of patient involvement in planning and advising development of the registry was not afforded while VFQ-25 was adopted as the key PRO measurement.

A total of 15 SOPs were developed for procedures including CSV, data management, monitoring, audit, management of imaging data and others in order to ensure that operations are

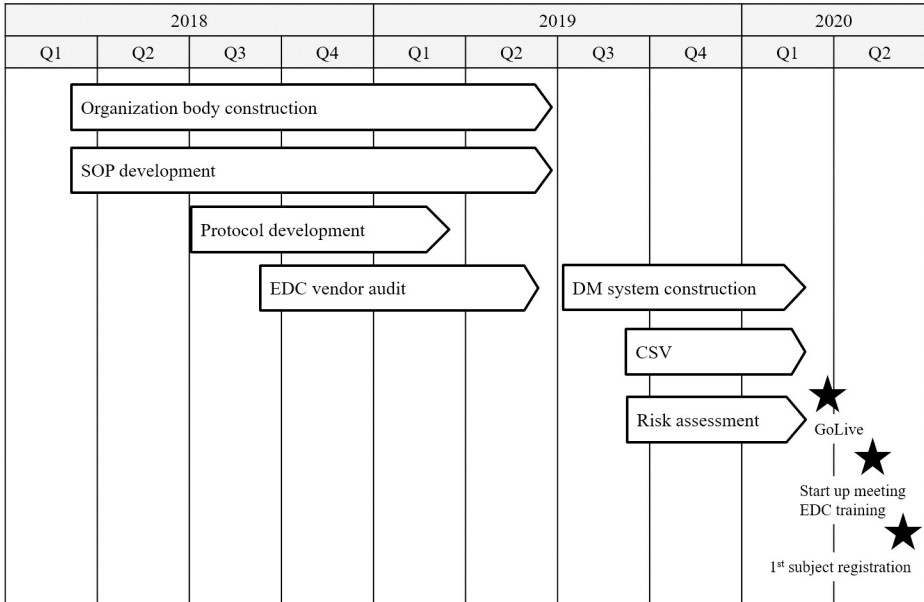

**Fig 3. Overall timeline for building the glaucoma registry.** Abbreviations: CSV, computerized system validation; DM, data management; EDC, electronic data capture; SOP, standard operating procedure.

carried out appropriately in compliance with the AHRQ publication and GAMP5 guidance (Table 1). The manual of the SOP management procedure (L1-THO-SOP-001) stipulated that a system owner (Prof. Toru Nakazawa of Tohoku University) was responsible for determining, developing and maintaining a set of relevant SOPs for operation of the registry. As stipulated

**Table 1. List of standard operating procedures for quality management systems.**

| Document ID | Document name | Date of issue |
|---|---|---|
| L1-THO-SOP-001 | SOP management procedure manual | 24-Dec-2019 |
| L1-THO-SOP-004 | Document management procedure manual | 24-Dec-2019 |
| L1-THO-SOP-002 | Risk assessment procedure manual | 5-Jul-2018 |
| L1-THO-SOP-003 | Vendor audit procedure manual | 5-Jul-2018 |
| L1-THO-SOP-005 | EDC user management manual | 24-Dec-2019 |
| L1-THO-SOP-006 | CSV policy | 24-Dec-2019 |
| L1-THO-SOP-007 | CSV procedure manual | 24-Dec-2019 |
| L1-THO-SOP-008 | Data management procedure manual | 21-Feb-2020 |
| L1-THO-MAN-001 | CRF data entry manual | 22-May-2020 |
| L1-THO-MAN-002 | EDC operation manual | 22-May-2020 |
| L1-THO-SOP-009 | Monitoring procedure manual | 1-Mar-2019 |
| L1-THO-MAN-003 | Risk assessment for risk-based monitoring manual | 11-Apr-2020 |
| L1-THO-SOP-010 | Audit procedure manual | 1-Mar-2019 |
| L1-THO-SOP-011 | Imaging data management procedure manual | 20-Sep-2019 |
| L1-THO-SOP-012 | Equipment quality control procedure manual | 20-Sep-2019 |

Abbreviations: CRF, case report form; CSV, computerized system validation; EDC, electronic data capture; SOP, standard operating procedure

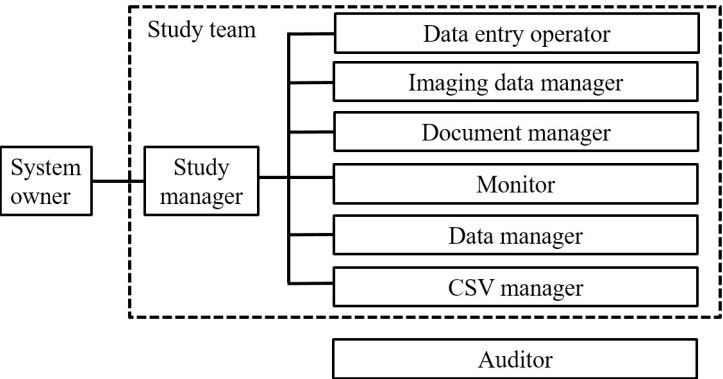

**Fig 4. Organization body for the glaucoma registry.** The system owner was fully responsible for owning and overseeing the registry. The study manager was responsible for conducting the study and managing the quality of the registry. The auditor was independent from the study team.

in the SOP of CSV policy (L1-THO-SOP-006), the system owner was assigned as ultimately responsible for the availability and maintenance of the computerized system, and fully responsible for owning and overseeing the registry. Two further personnel were then assigned: an SOP manager was assigned to manage and number the approved SOPs with a unique ID (L1-THO-SOP-004), and a study manager was assigned as responsible for conducting the study and managing the quality of the registry. The study manager additionally assigned a data entry operator (L1-THO-SOP-008), imaging data manager (L1-THO-SOP-011), document manager (L1-THO-SOP-004), auditor (L1-THO-SOP-010), monitor (L1-THO-SOP-009), data manager (L1-THO-SOP-008), and CSV manager (L1-THO-SOP-007), who were considered essential to ensuring that the registry's quality management system was implemented with regulatory grade quality (Fig 4). The qualification requirements and designation process for the managers of particular procedures were defined in each SOP.

The target subjects of the registry were set as Japanese glaucoma patients aged 20 years or older who met the eligibility criteria of the protocol. The target sample size was 3000 subjects, and the planned enrollment period was 5 years. The protocol scheduled that the timing of data capture of ophthalmology measurements should be every 4 months (visit window: +/- 1 month) after enrollment of the patient, and should include intraocular pressure, fundoscopy, stereo-fundoscopy, visual field, laser speckle flowgraphy (LSFG), optical coherence tomography (OCT), visual acuity. Further, refractometry, axial length, anterior segment OCT and administration of the QOL questionnaire of the 25-item National Eye Institute Visual Function Questionnaire (VFQ-25) should be scheduled every 12 months (visit window: +/- 1 month). Patient demographic data and medical history were collected at baseline (S1 File). The protocol for the registry was reviewed and approved by the ethical review committees of Tohoku University (Approval No. 2018-1-987) and Keio University (Approval No. 191114–1). An informed consent document was prepared based on the protocol and in compliance with local ethical guidelines for medical and biological research involving human subjects [25]. The designated investigators recruited the glaucoma patients in the routine care setting at the study sites. Minimal exclusion criteria were set to allow consecutive patient enrollment in the registry and the investigators registered the eligible patients after they had provided a written informed consent prior to their entry to the registry.

The data management plan consisted of data/paper flow and data management procedures developed in accordance with the data management procedure manual (SOP L1-THO-SOP-

008). The data/paper flow illustrated the sources, types, and specifications of the data and materials to be captured for the registry. The data fields of patient background (e.g. sex, birth date, type of glaucoma/diagnosed date, concomitant therapy, and medical history), vital signs, and ophthalmology measurements in the EHR were defined according to the study protocol and data management plan. The VFQ-25 questionnaire was administered to patients and collected in paper format. The data from these source documents were manually transcribed into the EDC by the data entry operators. Imaging data from the ophthalmology measurements were stored separately and linked with the Medrio EDC data via a unique registry ID (Fig 5). The data management procedures included data review by the data manager, query resolution, and data base lock and release.

The SOP of EDC user management manual (L1-THO-SOP-005) provided the policy and procedures to control data access for EDC users of the registry. The data access policy allowed only the designated EDC users to have access to the data captured in the field of the EDC according to the user's roles. For example, the investigators are able to only edit the data of their own site. The procedures for data-lock and export to generate research dataset was stipulated for investigations within the study team. We have not provided the SOP for provision of the registry data to investigators outside of the current research organization.

**Computerized system validation.** We decided that the data management system should be constructed based on Medrio EDC (R39.3), a cloud-based EDC platform (Medrio Inc., San Francisco, CA, US). Medrio EDC is 21 CFR Part 11-compliant, meaning that it is GCP-compliant and has an audit-trail function. The vendor was audited by EP-Techno Co., Ltd. via remote postal audit independently from the study team in accordance with the vendor audit procedure manual (SOP L1-THO-SOP-003). EP-Techno developed a list of questionnaires based on the GAMP5 and ERES guidelines [21, 22] and sent them to Medrio Inc. by postal mail. Medrio Inc. then completed the questionnaires and returned them to EP-Techno by mail with supporting documents. As the vendor audit reported that no items required

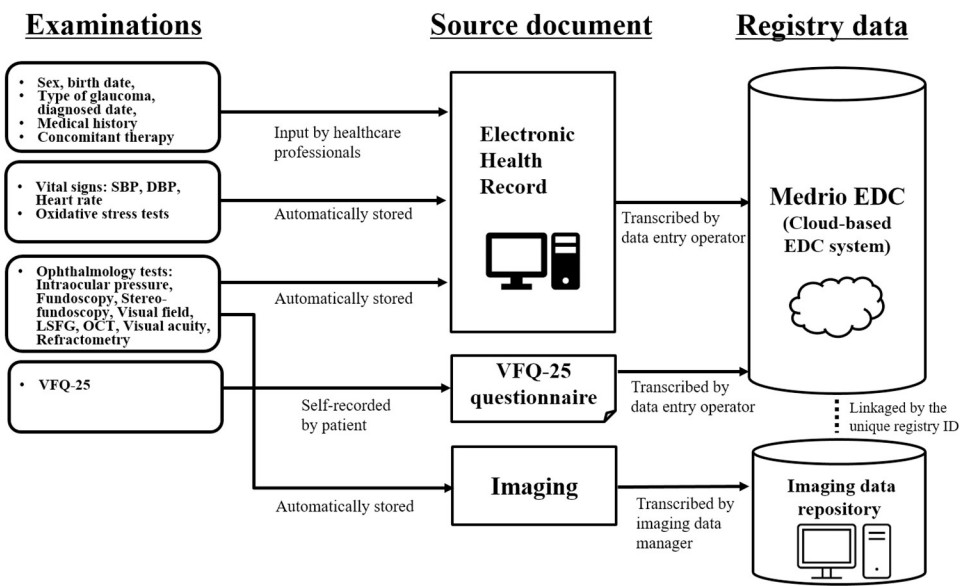

**Fig 5. Data flow diagram for the glaucoma registry.** Abbreviations: CRF, case report form; DBP, diastolic blood pressure; EDC, electronic data capture; HR, heart rate; LSFG, laser speckle flowgraphy; OCT, optical coherence tomography; SBP, systolic blood pressure; SOP, standard operating procedure; VFQ, Visual Function Questionnaire.

improvement, we finally confirmed the adoption of Medrio EDC as the EDC system for the glaucoma registry.

The V-model framework was used to validate the EDC system in the following 5 steps: 1) determine the user requirements specification, 2) determine the functional specification, 3) configuration, 4) functional testing, and 5) user acceptance testing (UAT) in accordance with the CSV procedure manual (SOP L1-THO-SOP-007). The user requirement specification was determined based on the data specifications according to the study protocol and data/paper flow. The functional specification described the data collection formats and structures while maintaining compatibility with the standards set by the Clinical Data Acquisition Standards Harmonization (CDASH) [26], except for examinations and diagnostic images specific for ophthalmology. The CSV manager configured the EDC system according to the functional specification. We referred to the definitions of the data elements and fields for ophthalmology tests used in existing clinical trials of glaucoma, considering that the data of the present registry might be used for an external control arm in clinical trials. A total of 183 data fields were configured as placeholders for study measurement. 91 automated edits (e.g. skip logic and range check) were configured to check errors in data entry into the EDC system. The designated validator within the study team executed the prespecified functional tests and user acceptance tests. The functional tests consisted of 11 test items to verify whether the configured EDC system functions as determined by the specifications (S1 Table). All 11 test items of the functional test were passed on January 15th 2020. Following this, the validator executed the UAT under a prespecified scenario consisting of 12 test items to verify the processes, including user account activation, data inputs, query solution, and report output (S1 Table). All 12 test items of the UAT were passed on January 23rd 2020. The validation report was completed on January 30th 2020, allowing the completion of CSV for the EDC system to be declared. This was followed by approval of the EDC system to begin operations (i.e., GoLive).

**Risk assessment and control.**   Informed consent, eligibility assessment and major ophthalmology measurements were determined as CTQ factors to achieve the goal of the registry in accordance with the risk assessment procedures described in the risk-based monitoring manual (SOP L1-THO-MAN-003). From the perspective of clinical relevance in evaluating glaucoma therapies, the major ophthalmology measurements included ocular pressure, visual field, visual acuity, and OCT. Quality standards were operationalized based on these CTQ factors as follows: for major ophthalmology measurements, "error and missing data are not detected after study participation"; for informed consent, "make sure that all study participants have provided informed consent through appropriate procedures"; and for eligibility assessment, "study participants meet all eligibility criteria stipulated in the protocol".

A total of 22 risk items were identified during the conduct of the study. These were subsequently classified into three risk categories according to the RACT method, namely as high (2 items), medium (13 items), and low (7 items) risk, after following the procedures in SOP L1-THO-MAN-003. Those risks which potentially affect the CTQ factors were classified as high risk, and included missing data in the major ophthalmology measurements without an appropriate medical reason (risk score: 12) and data entry errors in the major ophthalmology measurements (risk score: 12) (Table 2). The risk items rated 12 were related to the CTQ factors of major ophthalmology measurements.

Training was conducted at a start-up meeting held on July 5th 2020, with participation by 20 study staff of the registry. The study manager instructed participants about the registry's goal, eligibility criteria, and schedule of ophthalmology measurements in the protocol. The training session lasted about 1 hour. Study staff completed the SOP training for their relevant procedures by self-training. The first EDC user training was also held at the start-up meeting for data entry operators, investigators and monitors in accordance with the EDC user

**Table 2. Risk assessment.**

| ID | Risk | Category | Risk assessment | | | |
|----|------|----------|--------|-------------|---------------|------------|
| | | | Impact | Probability | Detectability | Risk score |
| 1 | Registration without consent | Informed consent | High | Low | Medium | 6 |
| 2 | Not obtaining consent in the appropriate version | Informed consent | High | Low | Medium | 6 |
| 3 | No signature on consent document | Informed consent | High | Low | Medium | 6 |
| 4 | Lost signed consent document | Informed consent | High | Low | Medium | 6 |
| 5 | Mis-registration of patient | Registration | High | Low | Difficult | 9 |
| 6 | Not filling out list of registration | Registration | Low | Low | Medium | 2 |
| 7 | Delay in registration | Registration | Low | Medium | Easy | 2 |
| 8 | Registration of ineligible patient | Registration | High | Low | Medium | 6 |
| 9 | Lost to follow-up | Registration | High | Low | Easy | 3 |
| 10 | Missing major measurement without appropriate medical reasons | Examination | High | Medium | Medium | 12 |
| 11 | Missing non-major measurement without appropriate medical reasons | Examination | Low | Medium | Medium | 4 |
| 12 | Protocol deviation of measurement | Examination | Medium | Low | Difficult | 6 |
| 13 | Defects in equipment | Examination | Medium | Low | Difficult | 6 |
| 14 | Data entry errors for major measurement | Case report form | High | Medium | Medium | 12 |
| 15 | Data entry errors for non-major measurement | Case report form | Low | Medium | Medium | 4 |
| 16 | Delay in data entry | Case report form | Low | High | Easy | 3 |
| 17 | Delay in query response | Case report form | Low | High | Easy | 3 |
| 18 | Delay in storing imaging data | Imaging | Low | Medium | Medium | 4 |
| 19 | Inconsistencies in the source document | Source document | Medium | Low | Difficult | 6 |
| 20 | Lost or discarded source document | Source document | High | Low | Medium | 6 |
| 21 | Registration from non-approved hospital | Administrative procedure | High | Low | Easy | 3 |
| 22 | Not approved by the Ethics Committee for initiation of study | Administrative procedure | High | Low | Easy | 3 |

management manual (SOP L1-THO-SOP-005). Ad-hoc user training is performed when the manuals of CRF data entry or EDC operation are revised and when a new study staff joins, and as deemed necessary in regard to the issues found at regular monitoring. Further, the equipment quality control procedure manual (SOP L1-THO-SOP-012) requires quality control inspection of test equipment used in primary laboratory measurements included in risk items ID 10 to 13 at least every 6 months to prevent missing data due to equipment malfunction.

A monitoring plan for the registry was developed in accordance with SOPs L1-THO-SOP-008 "Data management procedure manual", L1-THO-SOP-009 "Monitoring procedure manual" and the study protocol. To reduce the chance of data errors in the major ophthalmology measurements and to ensure agreement of the data between the source documents and the EDC, the monitoring plan pre-specified regular on-site monitoring by a designated site monitor at least every 3 months to perform source data verification (SDV) for all key data fields, including ocular pressure, visual field, visual acuity, and OCT. Remote monitoring by the data manager was planned to monitor compliance to the protocol through the EDC system, especially focusing on errors, missing and inconsistency of the data at the frequency of at least every 3 months. The automated edits to detect missing data for these key measurements (ID 10, 14) were configured to support regular and effective remote monitoring. This frequency of monitoring was determined with consideration to the fact that patient visit interval for the registry is 4 months.

## Compliance with the recommendations of regulatory guidelines

The glaucoma registry met 15 of 16 check items in the FDA guidance document 'Use of Real-World Evidence to Support Regulatory Decision-Making for Medical Devices'; 22 of 27 check

**Table 3. Assessment summary of recommendations related to data quality addressed in the regulatory guidelines.**

| Regulatory guideline | Number of Recommendations | | | |
|---|---|---|---|---|
| | Total | Addressed in the registry | Not addressed in the registry | Not applicable to the registry |
| FDA- Use of Real-World Evidence to Support Regulatory Decision-Making for Medical Devices | 16 | 15 | 1 | 0 |
| EMA- Discussion paper: Use of patient disease registries for regulatory purposes—methodological and operational considerations | 27 | 22 | 1 | 4 |
| IMDRF- Tools for Assessing the Usability of Registries in Support of Regulatory Decision-Making | 15 | 10 | 2 | 3 |
| MHLW- Points to Consider for Ensuring the Reliability in Utilization of Registry Data for Applications | 13 | 10 | 2 | 1 |

Abbreviations: EMA, European Medicines Agency; FDA, Food and Drug Administration; IMDRF, International Medical Device Regulators Forum; MHLW, Ministry of Health, Labor and Welfare

items in the EMA discussion paper 'Use of Patient Disease Registries for Regulatory Purposes–Methodological and Operational Considerations'; 10 of 15 check items in the IMDRF technical document 'Tools for Assessing the Usability of Registries in Support of Regulatory Decision-Making'; and 10 of 13 check items in the MHLW notification 'Points to consider for Ensuring the Reliability in Utilization of Registry Data for Applications' (Table 3). After the exclusion of inapplicable recommendations, the glaucoma registry did not meet two recommendations common to these guidelines. The first missing item was a common definitional framework (i.e., data dictionary), which was recommended in all four guidelines. The present registry does not include data coding dictionaries for the collection of adverse health events and coding of ophthalmology test measurements. The second item was policy on securing transparency (i.e., to make information about registry operations public and readily accessible to any interested party), which was recommended in the IMDRF and MHLW guidelines. In the present registry, we have partially addressed the latter by stipulating a method for disclosing information and publication of the registry in the protocol, and have accordingly disclosed information on registry holders, funding source, and purpose of the registry via the University Hospital Medical Information Network Clinical Trials Registry (UMIN-CTR) (Study ID: UMIN000037627), a clinical trial registry which complies with the requirements of the International Committee of Medical Journal Editors (ICMJE) and is accessible in the public domain. Details of compliance with the check items in each guideline are shown in S2 Table.

## Patient enrollment in the present glaucoma registry

As of August 2022, a total 165 patients had been enrolled in the present glaucoma registry at two study sites (Tohoku University Hospital and Seiryo Eye Clinic). The glaucoma registry was initiated at Tohoku University Hospital in April 2018, and Seiryo Eye Clinic subsequently joined in May 2020. The study manager assessed that the addition of a new study site would not affect the quality of the registry, on the grounds that the new site (Seiryo Eye Clinic) had sufficiency and sustainability in equipment and study staff available for the registry. As monitors and data entry operators at Tohoku University who were familiar with the protocol and procedure of the registry were able to conduct study operations at Seiryo Eye Clinic under established regional cooperation efforts between these sites, additional protocol training and EDC user training were not required or held.

Additional minor post-launch changes in the quality management system included a direct data import function from EHR systems to Medrio EDC by bulk-upload, which is equipped as

a standard function of Medrio EDC. The change control process included the validation and assessment of the newly added functions, in accordance with the pre-determined CSV process of the SOP.

## Discussion

Here, we report the construction and operation of a registry for glaucoma, a common disease, with the particular aim of building a quality management system which ensures regulatory-grade data quality using a risk-based approach.

A priori establishment of registry governance during the planning stage is essential to ensuring the built-in quality of data management systems, as recommended by EMA and MHLW [5, 8]. In addition, the AHRQ "Registries for Evaluating Patient Outcomes: A User's Guide: 4th Edition" stresses the importance of governance, and notes that registry governance can take many forms depending on the purpose of the registry, and that the goal of governance is to provide a mechanism for individuals to work together to achieve the goals of registry [6]. We developed a governing organization for the registry in accordance with the recommendations of the AHRQ publication. This organization was established in the form of formal collaborative research among academic organizations. The advantage of collaboration among academic organizations is that different expertise and skills in clinical, RWD, and data management available within each organization can complement each other.

Prior development of SOPs is also critical to ensuring better governance of a registry and to implement good quality management, particularly when regulatory submission of the registry data is intended. The ICH E6 (R2) guideline stresses the importance of implementing and maintaining a quality control system through SOPs to ensure compliance with the protocol and GCP [9]. In addition, the EMA's discussion paper "Use of patient disease registries for regulatory purposes–methodological and operational considerations" recommends that SOPs and work instructions be developed and followed to assure the data quality of a patient registry [5]. However, no official guidance provides a list of essential SOPs specific for operating a registry. We therefore developed a total of 15 SOPs to construct and operate the quality management system of the registry, in accordance with the AHRQ publication and GAMP5 guidance. Our present report may therefore serve to provide a reference list of SOPs necessary for disease registries intended for regulatory submission.

We found that the glaucoma registry addressed most of the quality recommendations in the official guidelines issued by multiple health authorities, with the exception of two recommendations that were not met. The first exception concerned the use of a coding dictionary. The purpose of the registry is to capture the patient journey in glaucoma; it is neither a medical product registry nor designed to collect data about adverse drug reactions, both of which necessitate the use of a coding dictionary for analysis. Moreover, as the types of ophthalmology tests performed were fixed and their results were numeric, coding of test methods was not necessary. An automated coding function was accordingly not provided in the data management system for the glaucoma registry, considering cost-benefit balance. Instead, manual coding of complications and medications is planned using MedDRA and WHO Drug Dictionary for regulatory activities independently of the EDC system when an analysis dataset is created in accordance with the data management plan [27, 28].

The second exception concerned the establishment of a policy to ensure transparency. The AHRQ publication stresses the importance of transparency because it contributes to public and professional confidence in the scientific integrity and validity of registry processes [6]. The MHLW notification "Points to Consider for Ensuring the Reliability in Utilization of Registry Data for Applications" recommends that registry holders specify and publish policies on these

matters (e.g. conflict of interest, operation and management system of registry holders, funding of the registry, purpose of the registry, disclosure of data) to ensure transparency in the operation and management of the registry [8]. The IMDRF's publication "Tools for Assessing the Usability of Registries in Support of Regulatory Decision-Making" advises that transparency is enhanced through the establishment and continuous maintenance of a publicly accessible website [24]. However, as the present registry has partially addressed these regulatory guideline requirements, we have no plan to run a publicly accessible website for the purpose of this glaucoma registry.

Procedures for change control should be in place before the start of patient enrollment and available throughout operations, similarly to the management of changes required for healthcare database maintenance [29]. Registries are in general designed to collect long-term follow-up and outcome data in a defined manner, including a variety of PROs and test results, while maintaining data quality standards. On the other hand, registries often evolve as emerging data elements and new study sites are added and need to accommodate various foreseeable as well as unforeseeable changes with regard to operations and funding resources resulting from changes in the research ecosystem in which they operate. Change controls should also consider changes in IT environment, including version control of platform software such as EHRs and data storage systems at study sites, coding nomenclature, and data management systems. While maintaining the quality of data in registries is the highest priority, the above changes have the potential to jeopardize quality, especially in terms of the reproducibility and longitudinality of the data. For example, when new participating study sites are added to the registry, procedures for site selection and training should be standardized through SOPs to ensure that data collection and quality are maintained at the same level before the new site is initiated.

Of particular note, this is the first academia-initiated glaucoma registry aiming to ensure regulatory-grade data quality in Japan. All study procedures were documented a priori, and standardized by the documentation of construction and operations for the whole registry project from its initiation. The quality of the registry was designed to meet the project goal, namely to ensure a level of data quality suitable for regulatory use as a historical comparator for single-armed clinical trials with new therapeutic agents. We trust that this report will serve as a reference case for the future design and construction of registries of regulatory-grade quality, and expect that the regulatory use of RWD, including in disease registries, will be accelerated to eliminate unmet medical needs.

## Conclusions

We established a quality management system for a glaucoma registry to ensure that the registry's data quality was both suitable for regulatory purposes, and allowed the prospective collection of long-term follow-up data of glaucoma patients.

## Supporting information

**S1 Table. Contents of functional tests and user acceptance tests.**
(DOCX)

**S2 Table. Assessment of addressed recommendations related to data quality in the regulatory guidelines.**
(DOCX)

**S1 File. Protocol synopsis: The glaucoma patient registry.**
(DOCX)

## Acknowledgments

The authors are grateful to Mr. Mitsuhide Yoshida and Mr. Keiya Inoue for their contribution as monitor or validator for the quality management system of the registry. We also thank Dr. Guy Harris of DMC Corp. (www.dmed.co.jp) for his support with the writing of the manuscript.

## Author Contributions

**Conceptualization:** Shinsuke Wada, Satoru Tsuda, Maiko Abe, Toru Nakazawa.

**Data curation:** Shinsuke Wada.

**Investigation:** Shinsuke Wada, Satoru Tsuda, Maiko Abe.

**Methodology:** Shinsuke Wada, Satoru Tsuda, Maiko Abe, Hisashi Urushihara.

**Project administration:** Shinsuke Wada, Satoru Tsuda.

**Resources:** Toru Nakazawa, Hisashi Urushihara.

**Supervision:** Toru Nakazawa, Hisashi Urushihara.

**Validation:** Satoru Tsuda, Hisashi Urushihara.

**Writing – original draft:** Shinsuke Wada, Hisashi Urushihara.

**Writing – review & editing:** Shinsuke Wada, Satoru Tsuda, Maiko Abe, Toru Nakazawa, Hisashi Urushihara.

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
