## [Decision Letter · Decision Letter 0]

20 Mar 2023

PONE-D-23-02739Design and implementation of a quality management system for a glaucoma registry to ensure the collection of data of regulatory-grade qualityPLOS ONE

Dear Dr. Urushihara,

Thank you for submitting your manuscript to PLOS ONE. After careful consideration, we feel that it has merit but does not fully meet PLOS ONE’s publication criteria as it currently stands. Therefore, we invite you to submit a revised version of the manuscript that addresses the points raised during the review process.

Please submit your revised manuscript within May 04 2023 11:59PM. If you will need more time than this to complete your revisions, please reply to this message or contact the journal office at plosone@plos.org. Please include the following items when submitting your revised manuscript:A rebuttal letter that responds to each point raised by the academic editor and reviewer(s). You should upload this letter as a separate file labeled 'Response to Reviewers'.A marked-up copy of your manuscript that highlights changes made to the original version. You should upload this as a separate file labeled 'Revised Manuscript with Track Changes'.An unmarked version of your revised paper without tracked changes. You should upload this as a separate file labeled 'Manuscript'.

We look forward to receiving your revised manuscript.

Kind regards,

Simon Grima, PhD

Academic Editor

PLOS ONE

Journal Requirements:

"I have read the journal's policy and the authors of this manuscript have the following competing interests: SW is an employee of Pfizer R&D Japan, and SW is also a graduate student at Keio University and contributed to the present study independent of Pfizer R&D Japan.

HU received research grants from Senju Pharmaceutical Co.,Ltd. The funder had no role in study design, data collection and analysis, decision to publish, or preparation of the manuscript.

The other authors have declared that no competing interests exist." 

Reviewers' comments:

Reviewer's Responses to Questions

**Comments to the Author**

1. Is the manuscript technically sound, and do the data support the conclusions?

Reviewer #1: Yes

Reviewer #2: No

2. Has the statistical analysis been performed appropriately and rigorously? 

Reviewer #1: N/A

Reviewer #2: No

3. Have the authors made all data underlying the findings in their manuscript fully available?

Reviewer #1: Yes

Reviewer #2: No

4. Is the manuscript presented in an intelligible fashion and written in standard English?

Reviewer #1: Yes

Reviewer #2: No

5. Review Comments to the Author

Reviewer #1: Thank you for submitting this manuscript which would be a useful addition to the literature that is accumulating on the quality assessment of disease registries that collect real world data. The focus of the work over here is to compare the quality of a registry against the criteria set by a number of international bodies for a registry to function as a suitable data resource for regulatory purposes. The authors show that their glaucoma registry ticks the majority of criteria and in the process provide a blue print that can be followed by other registries.

The manuscript would be further enhanced if the authors can consider the following points:

- can the authors provide further information on the quality assurance process - specifically, how often do they propose to perform source data checks and will those exercises as well as do these exercise need to be performed remotely or physically.

- the authors explain that the registry has protocols for data access but it seems that these are for those using the registry. They do not make it clear as to what is the protocol for data access by the wider group of investigators who may or may not be users of the registry. Its not clear as to what is the data access policy for research.

- on the point of governance, it is unclear as to who is in the steering committee and how is this managed

- data harmonisation - it seems that the registry has several uniques fields which are preclude it from data harmonisation; I think this needs some greater discussion; how were these fields created and who provided the guidance. Given that there may be several glaucoma registries in the world, did the investigators investigate the level of consensus amongst experts before embarking on the development of fields.

- it is unclear as to whether patients have been involved in advising on the contents

- training sessions - how often have these been planned for users

- I believe that the supplementary tables (esp table 2) are possibly more important than some of the figures which have been included in the main text as they constitute actual results.

Reviewer #2: 1. Please find a suitable title, because it is very poorly framed. It appears that the researcher doesn’t know how to encapsulate the entire idea in the title.

2. Reduce the overall length if the abstract and I am also not happy with the English. There is a need to improve style of writing. And also club the entire abstract and remove the headings given in the abstract.

3. The first line of the introduction is poorly written….. Disease/patient registries are……….

4. I am not even happy with the entire structure of introduction. The very important feature of an introduction is the background and the importance is missing.

5. RM is also very week and only focuses on curate clinical test and PRO findings in patients with 128 glaucoma and provide longitudinal data. How the three component approach will help in solving the problem. What is the sample design, the procedure for reaching to the target audience is also not clear.

6. Analysis, Findings conclusions are the weaker sections of this research work.

7. Kindly rewrite the entire paper

6. PLOS authors have the option to publish the peer review history of their article (what does this mean?). If published, this will include your full peer review and any attached files.

Reviewer #1: **Yes: **S. Faisal Ahmed

Reviewer #2: **Yes: **KIRAN SOOD

<quillbot-extension-portal></quillbot-extension-portal>

---

## [Author Response · Author response to Decision Letter 0]

11 May 2023

Reviewer #1 and #2: We have attached the response letter and incorporated your suggestions into the revision. They were very helpful. Thank you.

---

## [Editor Report · Decision Letter 1]

22 May 2023

A quality management system aiming to ensure regulatory-grade data quality in a glaucoma registry

PONE-D-23-02739R1

Dear Dr. Urushihara,

We’re pleased to inform you that your manuscript has been judged scientifically suitable for publication and will be formally accepted for publication once it meets all outstanding technical requirements.

Kind regards,

Simon Grima, PhD

Academic Editor

PLOS ONE
---

## [Editor Report · Acceptance letter]

25 May 2023

PONE-D-23-02739R1 

A quality management system aiming to ensure regulatory-grade data quality in a glaucoma registry 

Dear Dr. Urushihara:

I'm pleased to inform you that your manuscript has been deemed suitable for publication in PLOS ONE. Congratulations! Your manuscript is now with our production department. 

Kind regards, 

on behalf of

Professor Simon Grima 

Academic Editor

PLOS ONE